# Biological Activities of *Euphorbia peplus* Leaves Ethanolic Extract and the Extract Fabricated Gold Nanoparticles (AuNPs)

**DOI:** 10.3390/molecules24071431

**Published:** 2019-04-11

**Authors:** Hamed A. Ghramh, Khalid Ali Khan, Essam H. Ibrahim

**Affiliations:** 1Research Center for Advanced Materials Science (RCAMS), King Khalid University, P.O. Box 9004, Abha 61413, Saudi Arabia; hamedsa@hotmail.com; 2Unit of Bee Research and Honey Production, Faculty of Science, King Khalid University, P.O. Box 9004, Abha 61413, Saudi Arabia; essamebrahim@hotmail.com; 3Biology Department, Faculty of Science, King Khalid University, P.O. Box 9004, Abha 61413, Saudi Arabia; 4Blood Products Quality Control and Research Department, National Organization for Research and Control of Biologicals, Cairo 12611, Egypt

**Keywords:** *Euphorbia peplus*, nanoparticles, anticancer activity, antimicrobial activity, hemolytic activity, cytotoxic activity, insecticidal potential

## Abstract

*Euphorbia peplus* leaves extract (EpExt) and gold nanoparticles (AuNPs) phytofabricated with extract (EpExt-AuNPs) were investigated for biological activities. EpExt and EpExt-AuNPs were screened for: (i) anticancer activity against Hela and HepG2 cell lines; (ii) antimicrobial activity; (iii) hemolytic activity; (iv) cytotoxic or stimulatory effects; and (v) insecticidal activity. AuNPs (size 50 nm) were synthesized. (i) EpExt had a stimulatory effect (51.04%) on Hela cells and an inhibitory effect (−12.83%) on HepG2 cells while EpExt-AuNPs showed inhibitory effects (−54.25% and −59.64% on Hela and HepG2 cells respectively). (ii) Antimicrobial activity of EpExt-AuNPs was significantly higher (ranged from 11.67 mm to 14.33 mm) than that of EpExt (ranged from 5.33 mm to 6.33 mm). (iii) Both EpExt and EpExt-AuNPs displayed 100% hemolysis. (iv) A dose-dependent inhibitory effect of EpExt was observed (ranged from −48.5% to −92.1%), which was greater than that of EpExt-AuNPs (ranged from −32.1% to −69.1%) (v) EpExt-AuNPs was more lethal against mosquito larvae with lethal concentration (LC_50_) value (202.692 ppm) compared to EpExt (1430.590 ppm). In conclusion, EpExt-AuNPs were inhibitory against HepG2 and Hela cells, while EpExt inhibited HepG2 but stimulated Hela cells. EpExt-AuNPs had antimicrobial effects. EpExt showed dose-dependent inhibitory effects on splenic cells. EpExt-AuNPs were lethal against mosquito larvae.

## 1. Introduction

Many plants are known to have medicinal and therapeutic properties because of their phytochemicals and have tremendous applications in pharmaceutical industry [1]. An effort to investigate different biological properties of a plant extract that is used for nanoparticles synthesis may be done to explore the collective result of the metal with plant extract. The genus *Euphorbia* is the largest genus in the family Euphorbiaceae and comprises a large group of plants with over 2000 species in the world [2]. It has grasped the human interest for centuries, as it has exceptional diversity and near cosmopolitan distribution [3,4].

Studies have proven that different parts (leaves, stems, roots, and flowers) of the plants belonging to Euphorbiaceae family are rich in medicinal phytochemicals including tannins, alkaloids, glycosides, flavonoids, and other phenolic compounds. *Euphorbia peplus* belongs to Euphorbiaceae family and is indigenous to Europe, North Africa, and western Asia. Fast dividing human tissue are sensitive to its sap, which has long been utilized as a conventional cure for general skin lesions [3,4]. Phytochemicals such as sterols, diterpenes, C- and O-Glucosides, triterpene alcohols, cerebrosides, dihyroflavonol 3-O-monoglycosides, rutin, quercetin, kaempferol and myricetininside are reported in this plant [5,6]. This triggered interest in using its plant extract for biogenic synthesis of nanostructures by exhausting the reducing capacity.

For nanoparticles, it would not be false to say “the smaller the size the greater its potential” [7]. Metal nanoparticles have stimulated considerable attention because of their distinctive chemical, physical, and biological qualities and high priorities in different areas of research for the last few decades [8]. All the chemical, physical, and biological attributes of nanoparticles are strongly influenced by their specific size and shape [9]. For example, gold nanoparticles completely differ in properties from their respective bulk [10]. Use of nanoparticles is increasing rapidly in various fields such as biomedicine, tissue culture, health care, environment, drug delivery, gene delivery, etc. [11]. Various chemical and physical methods have long been used by many researchers for synthesis of nanoparticles. All procedures are either expensive or possibly risky for the environment as they encompass toxic and hazardous chemicals liable for numerous biological threats [8]. However, the synthesis of nanoparticles using naturally occurring reagents (bacteria, fungi, and plant extracts) is a substitute for chemical and physical methods. Biogenesis of nanoparticles has advantages over other procedures because of its simplicity, low cost, eco-friendly, reproducible, and repeatable results [12,13]. In biogenesis, metal ions are reduced by means of biological systems and as a result nano-size particles are produced, which have noticeable potentials compared to their counterparts. Gold, silver, platinum, zinc, and some other metals have been used along with various plant extracts for biogenic production of nanoparticles [14,15,16]. Gold nanoparticles fabricated with various plant extracts are gaining more attention because of their high antibacterial activity and easy reduction of salts.

Plants are naturally bestowed with several phytochemicals that could be useful for the synthesis of nanoparticles and chemical reactions [17,18]. In this study, *E. peplus* leaves ethanol extract (EpExt) was used to phytofabricate gold nanoparticles (EpExt-AuNPs) and their characterization was carried out using different methods. It was observed that the synthesized gold nanoparticles modified the bioactivity of the plant extract. In addition, EpExt and EpExt-AuNPs anticancer effects against HepG2 and Hela cancer cell lines, antimicrobial activity against selected pathogens, hemolytic activity, effect on splenic cell proliferation, and insecticidal activity against mosquito (*Culex pipiens*) larvae were investigated.

## 2. Results

### 2.1. Characterization of AuNPs

#### 2.1.1. Change in Color

The AuNPs synthesis was initially confirmed through the color alteration of the reaction mixture from off-white to dark brown (Figure 1).

#### 2.1.2. UV–Vis Spectrometry

EpExt-AuNPs were examined using UV-visible spectroscopy after the extract was mixed with an aqueous solution of HAuCl_4_ 3 H_2_O. UV-Vis spectra of plant leaves extract did not show any indication of absorption peak within 300–700 nm, while the plant extract mixed with HAuCl_4_ 3 H_2_O showed a distinctive peak absorption at 534 nm (Figure 2).

#### 2.1.3. FT-IR of EpExt and EpExt-AuNPs

FT-IR spectroscopy analyses were performed to recognize the bioactive components acquired from leaves extract to stabilize AuNPs using EpExt. FT-IR spectra (Figure 3) showed probable functional groups of *E. peplus*. FT-IR of EpExt (Figure 3A) was not significantly different from that of EpExt-AuNPs (Figure 3B), and might include alcohols, alkyne, and ketone. C=O was attributed to ketone not carboxylic acid because the peak of carboxylic acid had a broad shoulder at the region 2400–3500 cm^−1^ but the observed band appeared in the region 3200-3400 cm^−1^ that is specific to OH of alcohols not acids. C=O was not attributed to ketone, ester or aldehyde because there was no CH band. The prepared nanoparticles showed a medium broad band observed around 3200–3400 cm^−1^ that may be due to the occurrence of bonded O–H stretching of alcohol. Weak absorption around 2140–2100 cm^−1^ displayed the existence of stretching C≡C of alkyne. The very strong absorption band appearing at 1720 cm^−1^ was due to C=O stretching of ketone but not to ester or aldehyde because there was no CH band.

#### 2.1.4. Morphological Characterization Using Scanning Electron Microscopy (SEM)

Size and shape of gold nanoparticles (Figure 4) were characterized through SEM. The detailed analyses clearly showed the uniform spherical shape and low dimensions of AuNPs. Figure 4, which is magnified 10,000×, shows that the size of NPs was around 50 nm with interparticle distance.

#### 2.1.5. XRD Analysis of AuNPs Phytofabricated by EpExt

The conformation of prepared AuNPs was investigated by an X-ray diffraction (XRD) technique, and corresponding XRD patterns are shown in Figure 5. Gold nanocrystals exhibited four distinct peaks at 2θ = 38.2, 44.4, 64.5, and 76.4. The four peaks corresponded to standard Bragg reflections (111), (200), (220), and (311), respectively, of face center cubic (fcc) lattice plan according to JCPDS No. 04-0784.

### 2.2. Effects of EpExt and EpExt-AuNPs on Hela and HepG2 Cancer Cell Lines

EpExt showed a significant (*p* ≤ 0.05) stimulatory effect on Hela cells and a significant (*p* ≤ 0.05) inhibitory effect on HepG2 cells (Figure 6). EpExt-AuNPs showed significant (*p* ≤ 0.05) inhibitory effects on both Hela and HepG2 cell lines.

### 2.3. Antimicrobial Activity

Antimicrobial activity (in vitro) of EpExt, EpExt-AuNPs, HAuCl_4_ 3 H_2_O (1mM), and control are presented in Table 1. EpExt did not exhibit promising antimicrobial activity, while EpExt-AuNPs inhibited the microbial pathogen growth. Average diameters of zone of inhibition (ZOI) produced by the EpExt-AuNPs were larger (minimum 11.67 mm in *E. coli* and maximum 14.33 mm in *P. mirabilis*) than that of produced by EpExt (minimum 5.33 mm in *C. albicans* and maximum 6.33 mm in *Shigella flexneri*). Similarly, the average diameters of ZOI produced by HAuCl_4_ 3 H_2_O (1mM) were also large (minimum 6.33 mm in *C. albicans* and maximum 14.33 mm in *S. aureus*). The diameters of average ZOI produced by control were observed as 24.33 mm (maximum) in *C. albicans* and 18.67 mm (minimum) in *Shigella flexneri*. There was significant (*p* ≤ 0.05) difference between ZOI formed by EpExt, EpExt-AuNPs, HAuCl_4_ 3 H_2_O (1 mM), and control against all selected microbial pathogens. EpExt-AuNPs and HAuCl_4_ 3 H_2_O exhibited comparable (*p* ≤ 0.05) antimicrobial activity against *E. coli*, *S. aureus*, and *Shigella flexneri* but there was significant (*p* ≤ 0.05) difference between them against *P. mirabilis* and *C. albicans*.

### 2.4. The Lytic Effects of EpExt and EpExt-AuNPs on Red Blood Cells

The lysis percentage of the samples was assessed by comparing the absorbance of samples and the positive as well as negative controls. A significant difference was observed among the treatments (*p* ≤ 0.05). The positive control presented 100% lysis while the PBS alone displayed no lysis effect on RBCs. EpExt and EpExt-AuNPs displayed 100% lytic effect on RBCs (Table 2 and Figure 7).

### 2.5. Effects of EpExt and EpExt-AuNPs on Rat Splenic Cell Proliferation

The cytotoxic or stimulatory properties that may be found in the EpExt and EpExt-AuNPs were tested against normal rat splenic cells at applied concentrations. The results exhibited that there were significant (*p* ≤ 0.05) inhibitory effects between EpExt and EpExt-AuNPs at various concentrations (Table 3). These inhibitory effects were dose-dependent, where it decreased with the decrease in the extract concentration.

### 2.6. Effects of EpExt and EpExt-AuNPs on Culex pipiens Larvae

The larvicidal activity of EpExt-AuNPs was compared to that of an EpExt. Their effective larvicidal activities against the fourth instar larvae of *Cx. pipiens* after 24 h exposure times are shown in Table 4 and Figure 8. In general, 6.04–82.15% and 0.0–92.23% larval mortality was observed when the fourth instar larvae of *Cx. pipiens* were treated with the effective concentrations of EpExt (0–2500 ppm) and EpExt-AuNPs (0–500 ppm), respectively. The results reveal a substantial positive correlation between applied concentrations and mosquito larval mortality percentage in response to the concentrations EpExt and EpExt-AuNPs (*p* ≤ 0.05). Taking LC_50_ (concentration to kill 50% of larvae) value into consideration, EpExt-AuNPs (LC_50v_ = 202.692 ppm) proved to be more effective when compared to that of EpExt (LC_50_ = 1430.590 ppm) by about seven folds.

## 3. Discussion

The distinctive light brown color EpExt after addition of HAuCl_4_ 3 H_2_O can be credited to the excitation of surface plasmon response (SPR) through AuNPs, as confirmed in [19]. Some chemical compound (alkaloids, flavonoids, saponins, and steroids) may alter the color of solution as they act as a reducing agent for gold ions to a gold atom with the help of reductase enzyme. These specific enzymes discharged into a mixture can reduce the HAuCl_4_ 3 H_2_O to AuNPs by the capping agents like proteins [20].

It is actually recognized that UV-Vis spectroscopy could be used to examine the size and the shape of controlled nanoparticles in aqueous separation [21]. Figure 2 shows the UV-Vis spectra recorded from the reaction medium after 24 h. Absorption spectra of gold nanoparticles formed in the reaction media had a maximum absorbance peak at 561 nm, broadening of peak indicated that the particles were polydispersed. Bio reduction of gold ions through biomolecules in the plant leaves extract was the likely reason for this observation [22,23]. The particular shape and size of synthesized AuNPs can be examined by computing the XRD. The XRD spectrum measured in this study resulted in four intense peaks observed in the spectrum, which agrees to the Bragg’s reflection of AuNPs. The FTIR analysis of EpExt revealed the presence of alcohols, alkyne, and ketone. The less polar fractions of the latex of *E. peplus* were found to contain obtusifoliol, cycloartenol, 24-methylenecycloartanol, lanosterol, and 24-methylenelanosterol in the free and esterified triterpene alcohol fractions; 9-cis-tricosene as the major component of the hydrocarbon fraction; angelic acid (monocarboxylic unsaturated organic acid); and a new acyclic triterpene alcohol named peplusol [24]. The biological activities for nanoparticles were due to the morphology of the NPs and their sizes. The inhibitory effects of EpExt might be due the presence of the compound ingenol mebutate, which is a growth inhibitory mediator [25,26] in the extract. The inhibitory effect of the ingenol mebutate in the extract was not effective in the case of Hela cell line treatment. Gold nanoparticles might have entered the cells through endocytosis [27] leading to cascade of reactions that terminated with cell death.

These results of antimicrobial activities in accordance with those in [28,29]. The antimicrobial activity of EpExt-AuNPs might be due to their small size and uniform distribution. The ultra-small size of nanoparticles was responsible for deep penetration and AuNPs diffused through the cell membrane of microbial pathogens and disrupted the cell function causing cell death. Further, antimicrobial activity was also boosted by reactive oxygen species (ROS) generation and surface potential [30]. Previously, many types and sizes nanoparticles have been effectively used to induce the delivery of therapeutic agents [22,31] to cure infections in skin [32]. In addition, nanoparticles are used to prevent colonization bacteria on the surface of medical devices and as an antibacterial agent in food and clothing industries [33]. Due to its unique and nearly known mode of action and its antimicrobial properties against Gram-positive and -negative bacteria, the needs for developing a new generation of antibiotics make nanoparticles an alternative to traditional antibiotics to bypass the problem of drug resistance.

Biocompatibility and biological safety are the primary concern for any material to be used in cellular systems. Exogenous materials interact with various cellular systems and may lead to cell damage. The purpose of hemolytic activity measurement was to evaluate the biological safety of studied materials. The lysis effect of EpExt-AuNPs may be due to the direct effect of the nanoparticles on the membranes of the RBCs. Others researchers referred to the hemolytic effects as due to the induced release of oxidative stress products following exposure [34]. Hemolytic activity of EpExt and EpExt-AuNPs showed that both were very toxic. High hemolytic toxicity was observed in EpEx. It could be due to some phytochemicals. Bigoniya et al. [35] investigated the hemolytic activity of saponin isolated from *E. neriifolia* leaf (that belongs to the same genus as *E. peplus*). They reported high hemolytic activity of the saponins. In their study, silymarin also showed 97.05% hemolysis compared with triton which showed 100% hemolysis. The results of EpExt were in accordance with Kviecinski et al. [36] who evaluated the hemolytic toxicity of a latex extract from *E. tirucalli* and observed that aqueous ethanol extract of *E. tirucalli* caused high hemolysis. Similarly, Oliveira et al. [37] investigated the extracts of different aerial parts of 12 Amazonian plant species and reported that three extracts among them exhibited significant hemolytic activity. Many compounds derived from plants are responsible for in vitro hemolysis [38]. The results of high hemolytic toxicity of EpExt-AuNPs were comparable with those in [39]. The authors reported that the toxicity of silver nanoparticles was fairly size and dose dependent. The toxicity induced by small-sized AgNPs were attributed to the direct interaction of the NPs with the RBCs, resulting in the production of oxidative stress, membrane injury, and subsequent hemolysis. The potential application of both EpExt and EpExt-AuNPs may be achieved topically and then their high hemolytic activity might not be problematic. *E. peplus* is known to have the compound ingenol mebutate, which is a growth inhibitory mediator [25,26]. The decrease of inhibitory effects of EpExt-AuNPs might be due to masking of some active ingredients by nanoparticles. The precise larvicidal mechanisms of AuNPs are unidentified. It is expected that, because of their tiny size, the AuNPs infiltrate into the larva membrane and cause mortality. It is obvious from a regression equation that there is a linear correlation with AuNPs dose and mortality of mosquito larval instar, which justifies that higher quantities of AuNPs result in more intake of nanoparticles through the larval body and cause larval death accordingly.

It is reported that diffusion of NPs reduces the cell membrane permeability and reduction in ATP synthesis which finally disrupts the cellular function and cell death [40]. Additionally, once the AuNPs access the midgut epithelial membrane of larva, enzymes become inoperative, producing peroxide that leads to cell death [41,42].

It is also believed that penetration is stimulated by the attraction between Au+ and the cell membrane [43,44]. In the past, larvicidal capabilities of AuNPs synthesized by different plants have been described by many researchers. The authors of [45] reported the larvicidal activity of an aqueous leaf extract of *Hibiscus rosasinensis* against the mosquito (*Aedes albopictus*) larvae and the authors of [46] found that nanoparticles from fungus are effective against *Anopheles stephensi*.

## 4. Materials and Methods

### 4.1. Plant Extract Preparation

*E. peplus* leaves were collected in September 2017 from Abha, Saudi Arabia. Plant identification was performed by a plant taxonomist at King Khalid University. Tap water was used to clean the leaves many times with the aim to eliminate dust and soil particles from their surfaces. Afterwards, the leaves were cleaned with distilled water and desiccated in a shade at room temperature. Dry leaves were pulverized into powder using an electric blender [47,48]. To prepare the plant extract, soxhlet apparatus (Shiva Sci. Glass Pvt. Ltd. Gujarat, India) was used and 300 mL of 70% ethanol was added to 50 g of leaf powder. This solution was filtrated using a Whatman filter paper (Merck, Darmstadt, Germany) and flow-throw was air dried by rotary evaporator (SP Industries, Warminster, PA, USA) at 42 °C for 1–2 h. Flow-through dehydration brought about 2 g crude extract in semi-solid state. One gram of this dry substrate was liquefied in 100 mL of 70% acetone (Merck, Darmstadt, Germany) to obtain 1% stock solution.

### 4.2. Biosynthesis of Gold Nanoparticles (AuNPs)

AuNPs were synthesized by following [49] with some modifications. Briefly, 1 mL of 1 mM tetrachloroauric (III) acid trihydrate (HAuCl_4_ 3H_2_O) (Merck, Darmstadt, Germany) solution was mixed with 99 mL 1% stock solution of *E. peplus* extracts. Solution was kept at room temperature and stirring was continued until its color is transformed into brown that represent the development of nanoparticles.

### 4.3. Characterization of Gold Nanoparticles

Nanoparticles were subjected to UV-Vis spectra, at a wavelength range of 350–700 nm using UV-3600 Shimadzu spectrophotometer (Shimadzu Corporation, Kyoto, Japan) with a resolution of 1 nm according to the authors of [50]. Their shape was described through the scanning electron microscope (SEM, JEM-1011, JEOL, Tokyo, Japan) with an accelerating voltage of 90 KV. The functional groups present in EpExt-AuNPs were examined by Perkin- Elmer Spectrum 2000 FTIR (in the range of 500–4000 cm^−1^) (PerkinElmer, Inc. Waltham, MA, USA) at a rate of 16 times and the clarity of 4 cm^−1^ and X-ray powder diffraction (XRD) measurement of the EpExt-AuNPs was recorded with 2θ value in the range of 20–80° with a scan rate of 1° per minute on fine layers of the corresponding liquid drops coated on a microscopic glass slide using X-ray diffractometer (Rigaku Cooperation, Tokyo, Japan) functioned at 40 KV and 30 mA with Cu Kα1 radiations.

### 4.4. Effects of EpExt and EpExt-AuNPs Against Hela and HepG2 Cancer Cell Lines

HepG2 and HeLa cell lines (Merck, Darmstadt, Germany) were cultured separately in DMEM (Merck, Darmstadt, Germany) accompanied by 10% fetal calf serum (Web Scientific, Cheshire, UK), penicillin/streptomycin (100 U/mL/100 mg/mL, Web Scientific, Cheshire, UK) and 2 mM L-glutamine (Web Scientific, Cheshire, UK) at a cell density of 5000 cells/well in 96-well plate (Corning Life Sciences, Oneonta, NY, USA). The plates were kept at 37 °C under 5% CO_2_ and 85% humidity for 24 h. The media were removed from each plate and replaced with 200 µL/well fresh media containing 100 µg/mL of EpExt or EpExt-AuNPs separately. Cells with media only served as control culture. The cultures were continued for more 24 h at the same conditions described above. The cell viability assay was done according to the authors of [49] using Vybrant^®^ MTT Cell Proliferation Assay Kit (Thermo Fisher Scientific, Waltham, MA, USA) with little modifications. The media in micro well plates were changed with 100 µL new culture medium and 10 µL of 12 mM MTT were pippeted into each well and left for 3 h. Then, 100 µL of 0.1% acidified sodium dodecyl sulphate (SDS) (Merck, Darmstadt, Germany) were added to each well and the absorbance reading was noted at 570 nm. The results were represented by the percentage of the control at completion of each incubation period.

### 4.5. Antimicrobial Activity of EpExt and EpExt-AuNPs

#### 4.5.1. Microbial Strains and Media

Microbial pathogens that were investigated in this study were obtained from the Microbiological laboratory, The Department of Biology, Faculty of Science, King Khalid University, Abha, Saudi Arabia. *Escherichia coli*, *Proteus mirabilis*, and *Shigella flexneri* are Gram-negative while *Staphylococcus aureus* are Gram-positive bacteria. The fungus used in the experiment was *Candida albicans*. The bacteria were handled through standard techniques as described in [51] and preserved at 4 °C on nutrient agar slants. Both nutrient agar and broth (HiMedia Laboratories Pvt. Ltd. Mumbai, India) were prepared as per manufacturer’s instructions.

#### 4.5.2. Well Diffusion Assay for Antimicrobial Activity

The antimicrobial potential of EpExt and EpExt-AuNPs was measured by agar well diffusion method. All the microbial pathogens were inoculated in 10 mL broth and kept under a shaking incubator (Sheldon Manufacturing, Inc. Cornelius, OR, USA) at 37 °C overnight. Agar plates were prepared in a laminar flow hood by pouring autoclaved nutrient agar in plates. Distal ends of sterile Pasture pipettes were used to make wells of about 5 mm diameter in each agar plate. Fifty microliters of each microbial suspension (~10^8^ colony forming unit (cfu)/mL) were pipetted and equally distributed with sterile cotton swap (Citotest Labware manufacturing Co. Ltd. Mainland, China) on a single agar plate. Thirty microliters of diluted EpExt and EpExt-AuNPs were pipetted aseptically into the wells on agar plates. Penicillin–Streptomycin (20 units:20 µg) solution was taken as positive control. All the petri plates were kept in an incubator (Nüve Sanayi Malzemeleri, Ankara, Turkey) at 37 °C for 24 h. The diameter of inhibition zones near the outer surface of the wells were determined by following the [52].

### 4.6. Lytic Effects of EpExt and EpExt-AuNPs on Red Blood Cells (RBCs)

The lytic effects of EpExt and EpExt-AuNPs were determined by following the methodology of [53] with few modifications. EpExt and EpExt-AuNPs (1 mg/mL) were prepared in sterilized phosphate buffered saline (PBS). Cow blood (10 mL) in 15 mL Falcon tubes was gently mixed and centrifuged for 15 min at 1000 × *g*. Resulting supernatant was poured off and RBCs were elucidated three times with PBS. RBCs were suspended in PBS to obtain hematocrit (10%). Then, 100 µL of ready EpExt or EpExt-AuNPs were mixed with 900 µL of hematocrit in Eppendorf tubes (Eppendorf Middle East and Africa FZ-LLC, Dubai, UAE) and left in an incubator for 45 min at 37 °C. Triton X-100 (1%) and PBS alone were positive and negative controls, respectively. All Eppendorf tubes were centrifuged for 10 min at 2000 rpm. Once the centrifugation was complete, the absorbance of the supernatants was measured at 576 nm (Lamda 25; PerkinElmer, Inc. Waltham, MA, USA).

### 4.7. In Vitro Effects of EpExt and EpExt-AuNPs on Splenic Cells Proliferation

#### Splenic Cells Culture Preparation

Single-cell splenic suspension was prepared from rat according to [49]. Briefly, spleen of anesthetized healthy adult male Sprague Dawley rat (225 g), kindly supplied by animal house found at King Khalid University, was homogenized to release splenocytes and single-cell suspension was prepared in DMEM high glucose culture medium containing 10% foetal calf serum. Cells were adjusted to 0.05 × 10^6^/mL. The cell culture was accomplished in a 96-well plate (Corning Life Sciences, Oneonta, NY, USA) by pipetting 100 μL of splenic cell suspension (5000 cells/well) and 100 μL of EpExt and EpExt-AuNPs at 200, 100, 50, and 25 μg/mL final concentrations. Plates with cell cultures were kept at 37 °C for 72 h in 5% CO_2_ humid chamber (Thermo Fisher Scientific, Waltham, MA, USA) [54]. Vybrant^®^ MTT Cell Proliferation Assay Kit (Thermo Fisher Scientific) was used by following manufacturer’s instructions to calorimetrically observe the change in cell numbers of various treated cells [55]. The results were represented as increase/decrease percentage of growth as presented by [53].

### 4.8. Insecticidal Effect of EpExt and EpExt-AuNPs on Mosquito (Culex pipiens) Larvae

#### 4.8.1. Collection of Mosquito (*Cx. pipiens*) Larvae

*Cx. pipiens* larvae were collected from various location and their species were identified through insect taxonomist at the Entomology Unit, The Department of Biology, King Khalid University KSA. These mosquito larvae were placed in plastic trays filled with tap water. Dog biscuits and yeast (3:1) were administered to the rearing larvae [56]. Fourth instar larvae were used in the experiment and collected after six days post-hatching.

#### 4.8.2. Larval Bioassay

Mosquito larval susceptibility was tested through the method developed by [57]. Various treatments were applied by exposing early fourth instar larvae to different concentrations of EpExt (0–2500 ppm) and EpExt-AuNPs (0–500 ppm) for 24 h, divided into several groups in glass beakers filled with 100 mL of tap water. There were five repeats and each repeat consisted of 20 larvae per concentration and a control (tap water alone) was kept in an environmental compartment at 27 °C with a photoperiod of 16 h:8 h light/dark cycle. A usual larval food was administered to the larvae during the whole experiments. Larval mortality was documented at 24 h post-treatment with EpExt and EpExt-AuNPs. Mortality of larvae was confirmed through their movement even probed with a fine needle at the siphon or cervical region.

### 4.9. Statistical Analysis

#### 4.9.1. Statistical Analysis of Antimicrobial Activity

Antimicrobial activity was measured in terms of ZOI of an average of three replicates along with standard deviation (SD). An analysis of variance (ANOVA) was performed through Statistix 8.1 (Analytical software, Tallahassee, FL, USA). All pairwise comparison of means was performed with Tukey’s Honest Significant Difference (HSD) test. Means differed at *p* ≤ 0.05 were measured as statistically significant.

#### 4.9.2. Statistical Analysis of Larvicidal Bioassay

Completely randomized design was applied in this study. ANOVA was evaluated and means were related by LSD at *p* ≤ 0.05 with an SAS (version 9.3, SAS Institute Inc., Cary, NC, USA). LC_50_ was determined through a probit analysis program. Confidence intervals (95%), values, and DF of the *χ*^2^ goodness of fit tests, and regression equations, were assessed through computerized log-probit analysis. Abbot’s formula was applied to correct the larval percentage mortality.

#### 4.9.3. Statistical Analysis of Biological Activities

In the case of biological activities, the data were the means of three replicates. Variances in concentrations of RcExt and RcExt-AuNPs were evaluated by ANOVA by applying SPSS (version 17, SPSS Inc., Chicago, IL, USA). Differences with *p* ≤ 0.05 were reflected to be significant.

## 5. Conclusions

The synthesis of AuNPs was confirmed by all applied characterization methods. Gold nanoparticles fabricated with *Euphorbia peplus* leaves ethanol extract (EpExt-AuNPs) showed inhibitory effects against HepG2 and Hela cells, while the *E. peplus* extract alone (EpExt) inhibited HepG2 but stimulated Hela cells. EpExt-AuNPs had more antimicrobial effects on tested pathogens than EpExt. Both the extract and EpExt-AuNPs were hemolytic. EpExt showed more dose-dependent inhibitory effects on splenic cells than EpExt-AuNPs. EpExt-AuNPs was more lethal against mosquito larvae compared to EpExt.

## Figures and Tables

**Figure 1 molecules-24-01431-f001:**
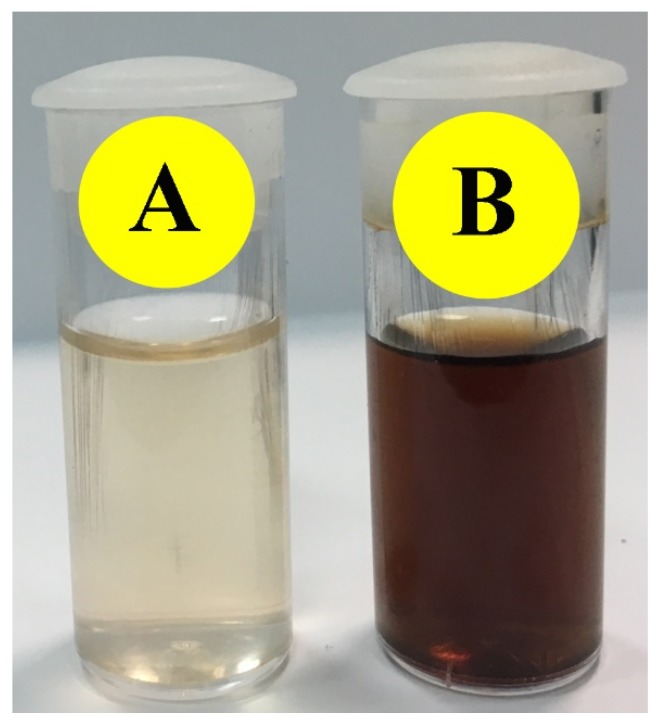
Confirmation of gold nanoparticles (AuNPs) synthesis through the change in color by the addition of tetrachloroauric (III) acid trihydrate (HAuCl_4_ 3 H_2_O) in *Euphorbia peplus* leaves ethanol extract (EpExt): before (**A**); and after (**B**).

**Figure 2 molecules-24-01431-f002:**
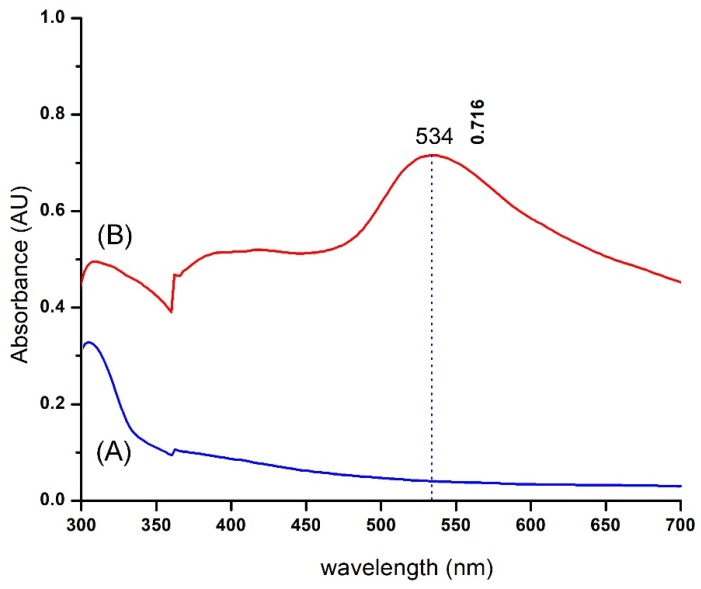
UV-Vis spectra of gold nanoparticles (AuNPs) formed through *Euphorbia peplus* leaves ethanol extract (EpExt): (**A**) EpExt; and (**B**) EpExt-AuNPs.

**Figure 3 molecules-24-01431-f003:**
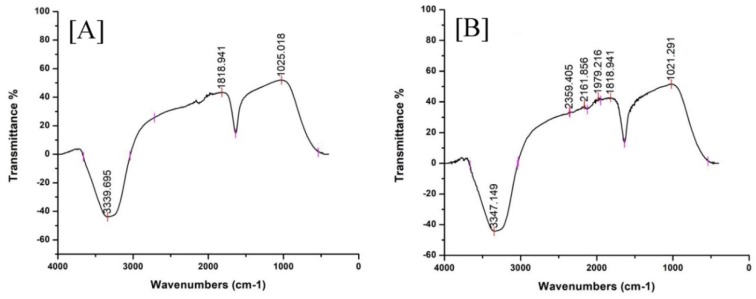
FT-IR spectra of: (**A**) *Euphorbia peplus* leaves ethanol extract (EpExt) alone; and (**B**) the AuNPs phytofabricated by EpExt.

**Figure 4 molecules-24-01431-f004:**
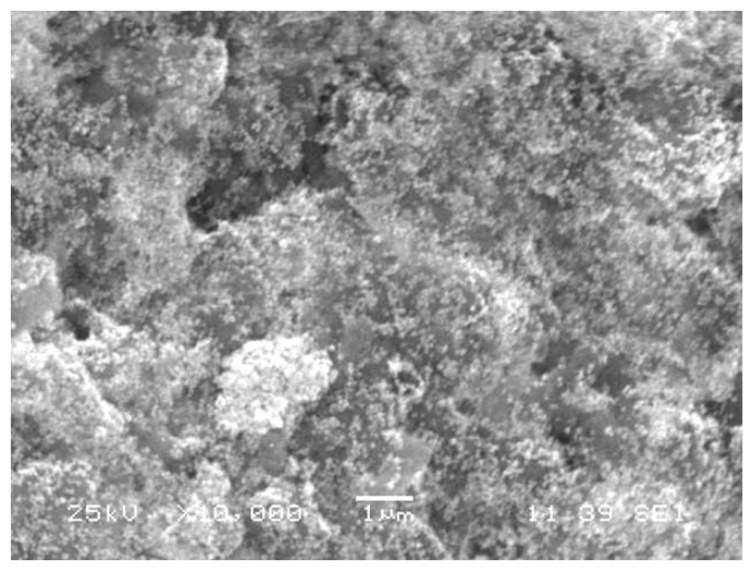
A scanned electron microscopic image of gold nanoparticles phytofabricated by *Euphorbia peplus* leaves ethanol extract, which are shown as crystalline, uniform, and aggregate.

**Figure 5 molecules-24-01431-f005:**
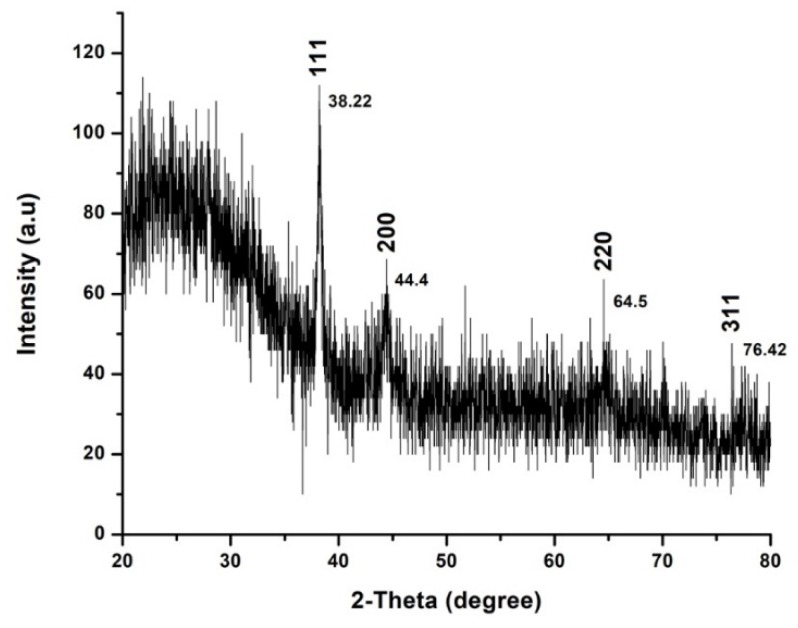
X-ray diffraction (XRD) pattern of gold nanoparticles (AuNPs) obtained using *Euphorbia peplus* leaves ethanol extract.

**Figure 6 molecules-24-01431-f006:**
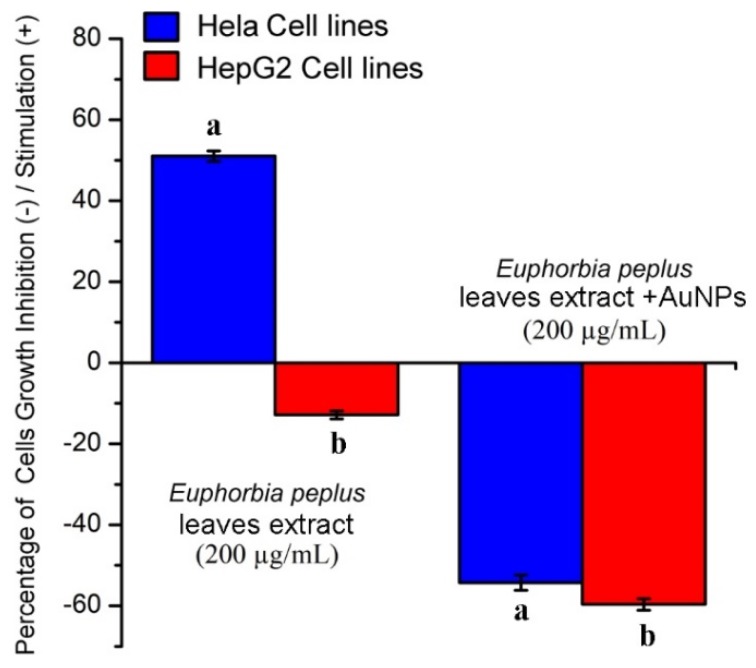
The effect of *Euphorbia peplus* leaves ethanol extract (EpExt) and EpExt-AuNPs on Hela and HepG2 cell lines growth. All data are presented as mean ± standard deviation (indicated as error bars) from three determinations. Bars with different letters in each group indicate significant (*p* ≤ 0.05) difference among the treatments.

**Figure 7 molecules-24-01431-f007:**
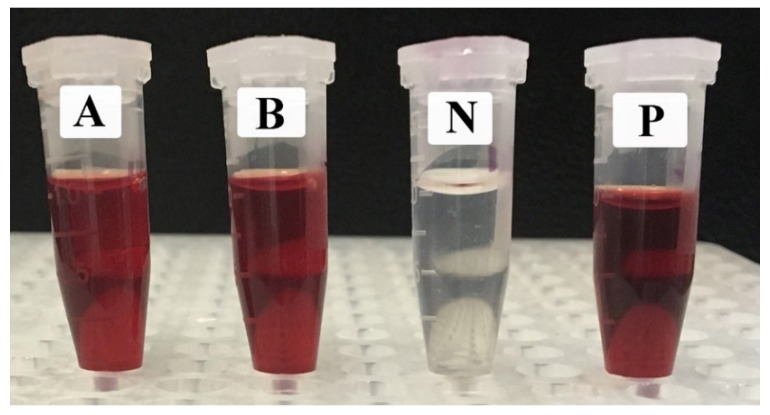
The hemolytic effect of *Euphorbia peplus* ethanol leaf extract (EpExt) (A) and extract with gold nanoparticles (EpExt-AuNPs) (B) on red blood cells. P and N are positive and negative controls, respectively.

**Figure 8 molecules-24-01431-f008:**
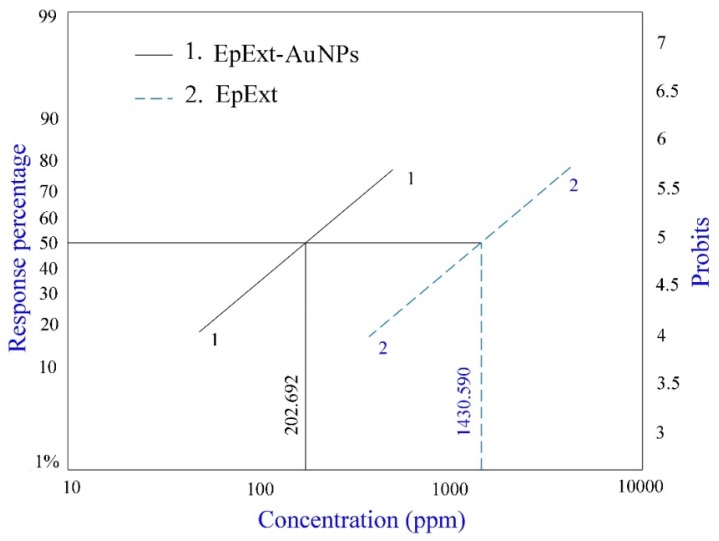
A relationship between the concentrations of *Euphorbia peplus* leaves extract (EpExt) and extract with gold nanoparticles (EpExt-AuNPs), and the fourth instar *Culex pipiens* larval mortality percentage. Line 1: EpExt – AuNPs; Line 2: EpExt.

**Table 1 molecules-24-01431-t001:** Antimicrobial potentials of *Euphorbia peplus* leaves ethanol extract, extract prepared gold nanoparticles (AuNPs), and tetrachloroauric (III) acid trihydrate (HAuCl_4_ 3 H_2_O) with 1mM concentration.

Treatments	Zones of Inhibition (mm)				
	*Escherichia coli*	*Proteus mirabilis*	*Staphylococcus aureus*	*Shigella flexneri*	*Candida albicans*
Plant extract	5.67 ± 1.15 ^c^	6.00 ± 1.73 ^c^	6.00 ± 1.00 ^c^	6.33 ± 0.58 ^c^	5.33 ± 0.58 ^c^
Extract + AuNPs	11.67 ± 1.57 ^b^	14.33 ± 1.53 ^b^	12.00 ± 1.00 ^b^	12.33 ± 1.15 ^b^	12.67 ± 0.58 ^b^
HAuCl_4_ 3 H_2_O (1mM)	12.67 ± 1.15 ^b^	6.67 ± 1.15 ^c^	14.33 ± 1.53 ^b^	12.67 ± 0.58 ^b^	6.33 ± 0.58 ^c^
Control	22.67 ± 1.53 ^a^	19.00 ± 1.00 ^a^	23.67 ± 1.53 ^a^	18.67 ± 1.53 ^a^	24.33 ± 2.08 ^a^

All data are expressed as mean ± standard deviation from three determinations. Superscript letters a, b, c, and d: Mean in each group not followed by same letters are significantly (*p* ≤ 0.05) different among the treatments.

**Table 2 molecules-24-01431-t002:** The lytic effect of *Euphorbia peplus* leaves ethanolic extract (EpExt) and extract with gold nanoparticles (EpExt-AuNPs) on red blood cells).

No.	Treatment	Absorbance at the Wavelength of 576 nm	Hemolysis (%)
1	EpExt	3.00 ± 0.000 ^a^	100 ± 0.00 ^a^
2	EpExt -AuNPs	3.00 ± 0.000 ^a^	100 ± 0.00 ^a^
3	Control (Negative)	0.0123 ± 0.002 ^b^	0 ± 0.00 ^b^
4	Control (Positive)	3.00 ± 0.000 ^a^	100 ± 0.00 ^a^

All data are represented as mean ± standard deviation (SD) from three determinations of each experiment. Superscript letters a and b: Mean in each group not followed by same letters are significantly (*p* ≤ 0.05) different among the treatments.

**Table 3 molecules-24-01431-t003:** Percent normal splenic cell growth stimulation/inhibition after treatment with different concentrations of *Euphorbia peplus* leaves extract (EpExt) and extract with gold nanoparticles (EpExt-AuNPs).

Concentration (µg/mL)	Percent (%) of Splenic Cells Growth Inhibition (−)/Stimulation (+)	
	EpExt	EpExt-AuNPs
25	−48.55 ± 0.45 ^b^	−32.64 ± 0.19 ^a^
50	−55.23 ± 1.06 ^b^	−32.11 ± 0.87 ^a^
100	−62.31 ± 0.58 ^b^	−48.31 ± 1.22 ^a^
200	−92.14 ± 1.55 ^b^	−69.06 ± 1.21 ^a^

All data are expressed as mean ± standard deviation from three determinations. Superscript letters a and b: Mean in each group not followed by same letters are significantly (*p* ≤ 0.05) different among the treatments.

**Table 4 molecules-24-01431-t004:** Susceptibility level of *Culex pipiens* to *Euphorbia peplus* leaves extract (EpExt) and extract with gold nanoparticles (EpExt-AuNPs) following continuous exposure for 24 h.

Bio Insecticide	Conc. (ppm)	Mortality (%)	LC_50_ (ppm)	LC_90_ (ppm)	Chi-Square	Slope	RRa
EpExt	500	24.96 ± 5.15 ^f^	1430.590	5805.463	18.97	3.073 ± 0.272	7.058
	1000	30.55 ± 6.52 ^e,f^					
	1500	40.68 ± 3.94 ^d,e^					
	2000	60.49 ± 8.19 ^b,c^					
	2500	82.15 ± 5.58 ^a^					
	0	6.04 ± 2.21 ^g^					
EpExt -AuNPs	100	20.33 ± 5.91 ^f^	202.692	529.463	3.684	2.107 ± 0.254	
	200	47.62 ± 10.18 ^c,d^					
	300	64.18 ± 3.88 ^b^					
	400	83.45 ± 5.69 ^a^					
	500	92.23 ± 4.81 ^a^					
	0	0.00 ± 0.00 ^g^					

Mortality percentage is expressed as mean ± standard deviation from five determinations. Superscript letters a, b, c, d, e, f, and g: Mean in each group not followed by same letters are significantly (*p* ≤ 0.05) different among different concentrations of EpExt and EpExt–AuNPs. Index compared with EpExt-AuNPs, Resistance Ratio (RR) compared with EpExt-AuNPs RR = larger LC50/smaller LC50; a = Index compared EpExt -AuNPs and EpExt used Resistance Ratio (RR).

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
