# Peer review of "Biological Activities of Euphorbia peplus Leaves Ethanolic Extract and the Extract Fabricated Gold Nanoparticles (AuNPs)"

_molecules, 2019, doi:10.3390/molecules24071431_

Reviewer 1 Report

All questions are answered.

Discussion about hemolytic activity is not still improved: "Discussion should be extended e.g. Hemolytic activity show that the extract and gold nanoparticles are very toxic. It is not appropriately discussed.”I think that authors should explain high hemolytic toxicity of studied samples. Does it have some influence on potential application of nanoparticles? E.g. Is it potential application by topic way and than is high hemolytic activity not problematic?

Author Response

Revised Manuscript ID. molecules- 475486 titled: " Biological Activities of Euphorbia peplus Leaves Ethanolic Extract and the Extract Fabricated Gold Nanoparticles (AuNPs)”

Thank you very for your email on 29 March 2019 regarding the resubmission of the above-mentioned manuscript after minor revision.

The revised manuscript is hereby uploaded in the journal online portal. We have replied about the suggestions and comments by the reviewers and also provided an English editing certificate.

If you need any further information about the manuscript, please feel free to contact me.

With best regards

Dr. Khalid A. Khan

(Corresponding author)

­­

RESPONSE TO REVIEWER COMMENTS

Q.1: Discussion about hemolytic activity is not still improved: "Discussion should be extended e.g. Hemolytic activity show that the extract and gold nanoparticles are very toxic. It is not appropriately discussed. “I think that authors should explain high hemolytic toxicity of studied samples. Does it have some influence on the potential application of nanoparticles? E.g. Is it a potential application by the topic way and then is a high hemolytic activity not problematic?

Response: In the revised manuscript, the required discussion part is now properly improved and written in detail along with some new references.

Reviewer 2 Report

The text needs soem language corrections, sometimes due to the editing, there are some parts left that make no sense like in line 48, there are the words "the Euphorbia genus " lost in the text , the complementing sentence has been deleted

Author Response

Revised Manuscript ID. molecules- 475486 titled: " Biological Activities of Euphorbia peplus Leaves Ethanolic Extract and the Extract Fabricated Gold Nanoparticles (AuNPs)”

Thank you very for your email on 29 March 2019 regarding the resubmission of the above- mentioned manuscript after minor revision.

The revised manuscript is hereby uploaded in the journal online portal. We have replied about the suggestions and comments by the reviewers and also provided an English editing certificate.

If you need any further information about the manuscript, please feel free to contact me.

With best regards

Dr. Khalid A. Khan

(Corresponding author)

RESPONSE TO REVIEWERS COMMENTS

Q.1: The text needs some language corrections, sometimes due to the editing, there are some parts left that make no sense like in line 48, there are the words "the Euphorbia genus " lost in the text, the complementing sentence has been deleted.

Response: The revised manuscript is now thoroughly checked for English language, grammar, and style (A certificate of English Editing is attached) by Prof. Philip Raines (Academic Organizer, Native English Language Section, King Khalid University, Abha, Saudi Arabia).

This manuscript is a resubmission of an earlier submission. The following is a list of the peer review reports and author responses from that submission.

Round  1

Reviewer 1 Report

The topic of the article is interesting. The scientific quality of the experiments design and results presentation should be improved.

I suggest to edit/rewrite 44-47 rows. It is not clear, in which rank this plant occupies third place?

2. Please, check and correct Latin names (italic font).

3. Fig.3 : FTIR spectrum of E. peplus should be provided.

4. XRD measurements should be done to confirm phase composition of the synthesized NPs.

5. Fig.5, 6, 7, and 8: Figure captions contain EpExt with HAuCl4.3H2O, whereas in corresponding text the abbreviation  EpExt-AuNPs  is used. So what was investigated in fact? Or Au NPs were not washed and used as prepared, in acetone dispersion?

6. Cytotoxicity study, Fig.5: the results for one concentration are shown. What about concentration dependence? Error bar should be given. The value Viability (%) is commonly used to estimate the results of cytotoxicity.

Reviewer 2 Report

tThe authors use plant extract to produce gold nanoparticles, a method that has been currently used . In the introduction and objectives it is not clear if they try to improve the bioactivity of the extract or  or if it just a tool  for the nanoparticles synthesis. 

Most of the figures presented have to be improved. The UV spectrum (Fig. 2) looks like it is a printscreen, there are no indication in the graph from which on is the extract and which is the  nanoparticle mix and there are some markings in one of the curves that are unreadable. The discussion about eh UV must be improved. Fig 4 must be changed  for a sharper  one that  presented  is blurred  one cannot see the particle shape.

In the FT-IR (figure 3) it is missing the extract spectra before the reaction, and the discussion must be improved, the representation of the triple bond in the discussion text is not correct. 

The figures for the antimicrobial, cytotoxicity and hemolysis can be replaced  for a table with the respective errors. 

In the experimental section the authors said that they performed statistical analysis, but it was not found in the results nor discussion. it must be presented to validate the results, mainly in the case of the insecticide assay, there no even the errors were presented. The significance of the results must be given.

In all assays, it was missing a control with the auric ions, in order to see that the effect was really due the nanoparticles.

Reviewer 3 Report

The manuscript of Ghramh H.A. et al. with title: “Biological activities of Euphorbia peplus leaves ethanolic extract and extract fabricated gold nanoparticles (AuNPs)” discussed synthesis, physicochemical properties and biological activities of extract and gold nanoparticles. The manuscript contains relative small amount of data. It discussed only one extract and one type of gold nanoparticles. The manuscript should be extend about more types of nanoparticles. Discussion should be extend. E.g. Hemolytic activity show that the extract and gold nanoparticles are very toxic. It is not appropriately discussed. I have also some remarks and question to manuscript:

Line 176-180. Authors discussed bands from IR spectra. The spectra should be more discussed. Does E. peplus contain some alkyne compounds? Is the band with wavenumber 1720 cm-1 C=O of carboxylic acid. Doesn’t it be band for other groups as aldehyde, ester, ketone…?

Line 186. Literature 26 and 27 discussed about silver nanoparticles. The results should be compared with activities of gold nanoparticles. There are large differences between gold and silver.

Manuscript contains several wrong citations Lines 169-170 and line 218

Latin names of organisms should be written by italic.

Sever subscripts and superscripts are missing

Sign Euphorbia peplus extract + HAuCl4.3HO is not correct. The mixture does not contain chloroauric acid.

Is the preparation of gold nanoparticles biosynthesis?